# A Model of a Zebrafish Avatar for Co-Clinical Trials

**DOI:** 10.3390/cancers12030677

**Published:** 2020-03-13

**Authors:** Alice Usai, Gregorio Di Franco, Patrizia Colucci, Luca Emanuele Pollina, Enrico Vasile, Niccola Funel, Matteo Palmeri, Luciana Dente, Alfredo Falcone, Luca Morelli, Vittoria Raffa

**Affiliations:** 1Department of Biology, University di Pisa, S.S. 12 Abetone e Brennero 4, 56127 Pisa, Italy; patrizia.colucci@student.unisi.it (P.C.); luciana.dente@unipi.it (L.D.); 2Department of Traslational Research and New Technologies in Medicine and Surgery, General Surgery Unit, University of Pisa, Via Paradisa 2, 56124 Pisa, Italy; gregorio.difranco@med.unipi.it (G.D.F.); niccola.funel@gmail.com (N.F.); matteo.palmeri@med.unipi.it (M.P.); luca.morelli@unipi.it (L.M.); 3Department of Surgical, Medical, Molecular Pathology and Critical Area, Division of Surgical Pathology, University of Pisa, Via Paradisa 2, 56124 Pisa, Italy; l.pollina@ao-pisa.toscana.it; 4Division of Medical Oncology, Pisa University Hospital, Via Roma 67, 56126 Pisa, Italy; envasile@tin.it (E.V.); alfredo.falcone@med.unipi.it (A.F.)

**Keywords:** patient-derived xenograft, zebrafish avatar, chemosensitivity, equivalent dose, translational research

## Abstract

Animal “avatars” and co-clinical trials are being developed for possible use in personalized medicine in oncology. In a co-clinical trial, the cancer cells of the patient’s tumor are xenotransplanted into the animal avatar for drug efficacy studies, and the data collected in the animal trial are used to plan the best drug treatment in the patient trial. Zebrafish have recently been proposed for implementing avatar models, however the lack of a general criterion for the chemotherapy dose conversion from humans to fish is a limitation in terms of conducting co-clinical trials. Here, we validate a simple, reliant and cost-effective avatar model based on the use of zebrafish embryos. By crossing data from safety and efficacy studies, we found a basic formula for estimating the equivalent dose for use in co-clinical trials which we validated in a clinical study enrolling 24 adult patients with solid cancers (XenoZ, NCT03668418).

## 1. Introduction

Precision medicine refers to the tailoring of medical treatment to the individual characteristics of each patient [1]. The “mouse avatar” is an emerging approach in precision medicine in oncology which has recently increased in importance [2]. It involves the xenotransplantation of cancer cells from the patient’s tumor sample into mouse models for use in drug efficacy studies. Mouse avatars are thus used to run “co-clinical trials” [3]. In a co-clinical trial, the patient and murine trials are conducted simultaneously, and the drug efficacy response of the mouse study provides data to plan the best drug treatment for the patient’s tumor [4].

The advantage of this approach is that each patient has his/her own tumor growing in an in vivo system, thereby facilitating the development of a personalized therapeutic approach. Today, companies are providing mouse avatar generation and drug testing services to patients which cost tens of thousands of dollars [5]. The high cost is directly associated with the time-consuming process involved and the need for strains [6]. Unfortunately, this means that avatars are a cutting-edge technology that few can afford, thus posing a serious threat to the equal right to health for everyone.

The use of zebrafish was recently proposed in order to make avatars available for every patient and thus to make the approach sustainable for National Healthcare Systems. Zebrafish cancer models overcome the drawbacks of xenografts in mice [7]. The zebrafish is highly fecund, develops rapidly and requires simple and inexpensive housing. Zebrafish embryos are transparent, enabling the engrafted cells to be imaged in vivo, and they have high permeability to small molecules such as drugs used for chemotherapy. Importantly, they have a low ethical impact when used in the larval stage from fecundation to 120 h post fertilization [8].

Zebrafish embryos used as a model for human cancer cell xenografts were first reported in 2005 [9]. Since then, the use of zebrafish in vivo models of xenotransplantation has increased considerably [10]. To date several human cancer cell lines e.g., melanoma, glioma, adenocarcinoma, breast, pancreas and prostate cancer cell lines [11], as well as fragments of human cancer tissues [12], have been tested in zebrafish as the engraftment host. Embryos provide a rejection-free permissive environment, where after injection the xenotransplanted human cancer cells rapidly proliferate, migrate, form masses and induce neo-angiogenesis [13]. Most importantly, zebrafish embryo xenografts provide similar chemo-sensitive responses as mouse xenografts [14].

However, in order to move forward in a new paradigm of co-clinical trials using zebrafish avatars, various critical aspects need to be solved. The key issue is related to the lack of an “equivalent dose” for translating the chemotherapy dosage used in humans to zebrafish embryos because the interspecies allometric approach for dose conversion from human to animal cannot be applied. The caveat is that chemotherapy drugs need to be administered in the fish water rather than injected as parenteral formulations. Therefore, drug safety and efficacy assessments are necessary to estimate the equivalent dose to administer [15].

The present study aims to fill the gap regarding the dose conversion between zebrafish embryos and humans. A safety study was carried out in zebrafish embryos by testing 10 different chemotherapy regimens used in cancer treatment, i.e., 5-Fluorouracil + Lederfolin + Oxaliplatin (FOLFOX), 5-Fluorouracil + Lederfolin + Irinotecan (FOLFIRI), 5-Fluorouracil + Lederfolin + Oxaliplatin + Irinotecan (FOLFOXIRI), 5-Fluorouracil + Cisplatin + Epirubicin (ECF), 5-Fluorouracil + Lederfolin + Oxaliplatin + Docetaxel (FLOT), Gemcitabine + Cisplatin (GEMCIS), Gemcitabine + nab-Paclitaxel (GEM/nab-P), Gemcitabine + Oxaliplatin (GEMOX), Gemcitabine (GEM), 5-Fluorouracil (5-FU).

Safety data were matched with the efficacy study carried out in zebrafish embryos xenotransplanted with HCT 116 and MIA PaCa-2 cancer cell lines. We found a general criterion for dose equivalence that was validated in zebrafish avatars receiving fresh tissue fragments taken from surgical specimens of patients who had undergone surgical operations for hepato-biliary-pancreatic cancer and gastro-intestinal cancer.

## 2. Results

### 2.1. Zebrafish Safety Study: Estimation of the Maximum Tolerated Dose (MTD)

A dose-response analysis to determine the effects of chemotherapy treatment on embryos was based on an evaluation of the phenotype resulting from the exposure (i.e., normal, aberrant and dead). The embryos stage will hereafter be referred to as days post fertilization (dpf). We exposed 2 dpf embryos to different concentrations of 10 different chemotherapy treatments (GEM, GEMOX, GEM/nab-P, GEMCIS, 5-FU, FOLFOX, FOLFIRI, FLOT, FOLFOXIRI, ECF) for 72 h (Figure 1).

For the multidrug combination chemotherapy (Appendix A), we mixed drugs at the mass ratio used for humans, i.e., GEMOX was 10:1 in Gemcitabine:Oxaliplatin; GEM/nab-P was 8:1 in Gemcitabine:nab-Paclitaxel; GEMCIS was 40:1 in Gemcitabine:Cisplatin; FOLFOX was 33:2.4:1 in 5-Fluorouracil:Lederfolin:Oxaliplatin; FOLFIRI was 15.6:1.1:1 in 5-Fluorouracil:Lederfolin:Irinotecan; FOLFOXIRI was 37.6:2.4:1.9:1 in 5-Fluorouracil:Lederfolin:Irinotecan:Oxaliplatin; FLOT was 52:4:1.7:1 in 5-Fluorouracil:Lederfolin:Oxaliplatin:Docetaxel; and ECF was 56:1.2:1 in 5-Fluorouracil:Cisplatin:Epirubicin.

Chemotherapy treatments induced death and a variety of malformations in the embryos, including yolk sac edema, pericardial edema and spine deformation. For all regimens, deviation from phenotype without defects (normal phenotype) increased with the increase in drug concentration. Linear regression analysis showed an excellent relationship between the linear or logarithmic concentration of the chemotherapy drug and the incidence of normal phenotype (R^2^ > 0.95; *p* < 0.05 for all the protocols tested) or the incidence of mortality (R^2^ > 0.87; *p* < 0.05 for any protocol tested), (Figure 1). For all the chemotherapy treatments, the dose that was lethal to 25% of the population (LD25) and the concentration at which 50% of the normal phenotype was inhibited (IC50) were determined (Table 1). Data were expressed in a dimensionless form, i.e., as EPC/LD25 or EPC/IC50 with EPC (Equivalent Plasma Concentration), given by:(1)$$EPC=\frac{M}{V}$$
where *M* is the total amount (mg) of chemotherapy administered to humans by the clinicians involved in the present study, and *V* (mL) is the mean volume of human blood (the *EPC* value for each regimen is given in Table 1). Notably, for multidrug combination chemotherapy, EPC/LD25 or EPC/IC50 are independent of the specific drug of the combination, as the mass ratio among drugs is always kept constant.

In the present study, we fixed the 75th percentiles of the box plots (Figure 2) as the maximum tolerated dose (MTD). Interestingly, this value was similar for the two conditions, i.e., 4.1 and 4.5 for EPC/IC50 and EPC/LD25, respectively. Consequently, the safety study established a MTD = max (4.1, 4.5) = 4.5 as the threshold for the determination of the Equivalent Dose (*ED*) required for an acceptable safety level for the zebrafish trial. In fact, exposure of embryos to the chemotherapy concentration corresponding to MTD = 4.5 led to a high survival rate (>75%) for all the protocols tested.

### 2.2. Zebrafish Efficacy Study: Estimation of the ED (Equivalent Dose)

For the estimation of the equivalent dose (*ED*), we conducted an in vivo efficacy study based on human cancer cell lines, whose chemosensitivity has already been characterized in the literature. Specifically, 2 dpf embryos were xenotransplanted with DiI-stained human colorectal carcinoma cell line (HCT 116) or human pancreatic carcinoma cell line (MIA PaCa-2) into the yolk sac. The hours/days after the end of the xenotransplant will hereafter be referred to as hours or days post injection (hpi or dpi, respectively). To confirm the presence of the xenograft, injected embryos were screened by fluorescence microscopy 2 hpi. The screened embryos were randomly distributed in a multi-well plate (1 embryo/well) and equally divided among groups (control and chemotherapy regimens).

In the absence of chemotherapy, the DiI-stained area showed a statistically significant increase over time (Figure 3, control group), and the block or inversion of this tendency was considered as a hallmark of the chemotherapy effect. In fact, we tested four chemotherapy regimens (5-FU, FOLFOX, FOLFOXIRI, FOLFIRI), which are the standard of care for the treatment of colorectal cancers, on HCT 116 cells xenotransplanted in 2 dpf embryos.

Depending on the value of the MTD, we tested human-to-fish dilution factors of *d.f.* > 4.5. First, a *d.f.* = 8 was tested however data showed a statistically significant increase in the DiI-stained area at 1 dpi and 2 dpi for all the regimens, suggesting the inefficacy of chemotherapy treatment at the *d.f.* used (Figure 3A). We therefore tested chemotherapy protocols at a higher concentration, corresponding to *d.f.* = 5. Interestingly, FOLFOXIRI was found to inhibit the increase in the stained area at 1 dpi and 2 dpi (*p* > 0.05), in contrast to the control, 5-FU, FOLFOX and FOLFIRI which showed a statistically significant progression (Figure 3B). The effect of FOLFOXIRI treatment was also confirmed by the quantification of apoptosis in xenotransplanted (DiI-positive) cells revealing a significant increase in pyknotic nuclei with respect to the control group (no chemotherapy drugs) (Figure 4).

The next step was to confirm the value of *d.f.* = 5 for dose equivalence by testing its efficacy on a different model, i.e., xenotransplanted embryos receiving MIA PaCa-2 cell line. In fact, we tested four chemotherapy regimens (GEM, GEMOX, GEM-Nab, FOLFOXIRI), which are the standard of care for the treatment of pancreatic cancers, on MIA PaCa-2 cells xenotransplanted in 2 dpf embryos. GEM and GEM-Nab-P proved to be the most efficient regimens, with no statistically significant increase in the Dil-stained area at 1 dpi and 2 dpi, in contrast to the control, GEMOX and FOLFOXIRI (Figure 3C).

In accordance with these data, the equivalent dose *ED* = *d.f.* = 5 was used to run the co-clinical trial described below.

### 2.3. Zebrafish Avatar

A total of six patients operated on for adenocarcinoma of the colon (*n* = 2), pancreatic ductal adenocarcinoma (*n* = 2) and gastric adenocarcinoma (*n* = 2) were enrolled in the study (NCT03668418) to create the zebrafish avatar model. In order to preserve the tumor microenvironment, we xenotransplanted fresh tissue fragments screened by the histopathology unit of the University Hospital of Pisa, by modifying the protocol published by Marques et al. [12].

Briefly, the tissue was DiI/DiO-stained, disaggregated using Dumont forceps (No.5) into a relative size of ½–¼ the size of the yolk, and xenotransplanted into the yolk of 2 dpf embryos. After transplantation, embryos were incubated for 2 h at 35 °C, then screened to check for the presence of the stained tissue and imaged at 2 hpi, 1 dpi and 2 dpi. As DiI/DiO staining is not an indicator of cell survival, we monitored cell engraftment at two days post-xenotransplantation (2 dpi) by histological analysis. Hoechst staining showed healthy cell nuclei with a small fraction of pyknotic nuclei (10.21 ± 1.70%, *n* = 3 patient samples, illustrative images provided in Figure 5 and Appendix A) and H&E staining showed the presence of cancer cells that have the typical round-shape morphology with large nuclei (Appendix A).

Interestingly, H&E staining performed on zebrafish embryos xenotransplanted with fragments of normal tissue taken from normal mucosa or pancreatic parenchyma of the surgical specimen did not show any cells with a typical cancer morphology, but exclusively cells with a typical fibroblast-like shape (Appendix A).

We also performed immunostaining against human IgG to highlight the presence of human cells in the zebrafish avatar, two days after the xenograft (2 dpi). First, the human antibody specificity was validated against HCT 116 cancer cell lines xenografted into zebrafish embryos (Figure 6A). The staining was then performed on patient biopsies before (Figure 6B) and after xenotransplantation (Figure 6C). Occasionally, we also detected cancer cell migration, as highlighted by DiI and Anti-Human IgG positive cells (Figure 6D).

We measured the size of the region of interest (ROI) corresponding to the DiI stained area at 2 hpi, 1 dpi and 2 dpi (Figure 7A3–C3). The mean size of the tumor mass area measured in each time point was normalized with respect to the 2 hpi time point (relative area). We found an increase in the stained area versus time in all cases, which was statistically significant at 2 dpi with respect to the time point 2 hpi for five out of six patient samples (83%, Figure 8). The measurement of the size of the relative stained area was thus fixed as the primary measure of the study.

### 2.4. Zebrafish Trial

A total of 24 adult patients with pancreatic cancer (*n* = 12), colon cancer (*n* = 8) and gastric cancers (*n* = 4) undergoing chemotherapy treatment were recruited for this part of the study. After surgery and histopathology screening, patient biopsies were xenotransplanted into 100 zebrafish embryos. The injected embryos were randomly allocated among five groups: four therapeutic options and one control group. Groups were exposed to all chemotherapy options, according to the cancer type, by dissolving the chemotherapy in fish water, according to the *ED* = 5.

Two days post treatment, the response of zebrafish xenografts to the chemotherapy options was analyzed by monitoring the ROI size at 2 hpi, 1 dpi and 2 dpi (Figure 7). The chemotherapy protocols tested were 5-FU, FOLFOX, FOLFIRI and FOLFOXIRI for colon cancer; GEM, GEMOX, GEM/nab-P and FOLFOXIRI for pancreatic cancer and FOLFOX, FOLFORI, FLOT and ECF for gastric cancer. We adapted the “Response Evaluation Criteria In Solid Tumors (RECIST)” to the fish trial by defining the partial response (PR, at least a 30% decrease in the relative stained area at 2 dpi/2 hpi, taking the relative stained area at 2 dpi/2 hpi of the control group as a reference) and complete response (CR, at least a 90% decrease in the relative stained area at 2 dpi/2 hpi, taking the relative stained area at 2 dpi/2 hpi of the control group as a reference) (Figure 9).

For patient-derived xenograft (PDX) of colon cancers we observed a PR in 62.5% of cases to FOLFOX, FOLFIRI and FOLFOXIRI however a less frequent response (37.5% of patient’s samples) to 5-FU. A CR was observed only in a limited number of patient’s sample (12.5%) and only to FOLFIRI chemotherapy. For PDX of pancreatic cancers, we observed a PR to GEM/nab-P (58.33 % of patient’s samples), GEM (50%), GEMOX (50%), a limited PR to FOLFOXIRI (33.33 %) but no CR was observed for any of the chemotherapy treatments. For PDX of gastric cancers we observed a high incidence of PR to FOLFIRI (100% of cases) but a low incidence of PR to FOLFOX, FLOT and ECF (25% of cases); we also observed a CR to FOLFIRI in one sample out of four.

Interestingly, a zebrafish avatar can be used to perform a chemosensitivity assessment on an individual patient basis. The chemosensitivity assays of four representative patients (C024, C031, P025 and P030) are shown in Figure 10. For each patient sample, the effectiveness of the chemotherapy was evaluated. The analysis was performed in terms of relative area (2 dpi/2 hpi), comparing the treated groups with respect to the control group. As with the two cases of colon cancer, we observed a significant response (*p* = 0.03) to FOLFOXIRI treatment in patient’s sample C024, and to 5-FU (*p* = 0.05) and FOLFIRI (*p* =0.02) in patient’s sample C031. Again, as with the two cases of pancreatic cancer, FOLFOXIRI proved to be the most efficient regimen in patient’s sample P025 (*p* = 0.02) and in patient’s sample P030 (*p* = 0.04).

## 3. Discussion

Many studies have demonstrated that preclinical models hold great promise for the implementation of personalized medicine strategies [16,17,18]. The mouse avatar has been proposed and implemented by several research groups, however mice still have important practical limitations [2,19,20]. The zebrafish offers distinct advantages over murine-based co-clinical trials because of the relatively simple, rapid and cost-effective method involved to establish a human tumor xenograft model [21]. The zebrafish avatar approach could be used for evaluating individual patient drug responses in a clinically relevant setting or for the high-throughput screening of new molecules. Considering the validity of the zebrafish avatar and the affordable cost involved, exploiting this model in clinical practice has become feasible. To do this, the equivalent dose conversion from human to fish needs to be identified.

In this work, we found a general dose conversion criterion based on the following formula:(2)$$Cfish=\frac{M}{V}/ED$$
where *c_fish_* (mg/mL) is the chemotherapy concentration in fish water, *M* is the total amount (mg) of chemotherapy administered to humans, *V* (mL) is the volume of human blood, and *ED* is the Equivalent Dose (dimensionless) which we estimated to be *ED* = 5. This involves performing a 5-fold dilution in the fish water, with respect to the EPC, to obtain effectiveness of the drug combination used for the co-clinical trial. We estimated this value by matching data collected from the safety and efficacy studies performed on zebrafish. The safety study was performed on WT embryos to establish the maximum tolerated dose for chemotherapy in fish embryos which was estimated to be MTD = 4.5. At this MTD, at least 75% of the population of treated embryos survived all the protocols tested.

The efficacy study was performed on embryos xenotransplanted with human cancer cell lines (colorectal cancer cell line HCT 116 and pancreatic cancer cell line MIA PaCa-2 cells) whose response to chemotherapy had already been characterized. Specifically, we found that HCT 116 responded to FOLFOXIRI treatment with the highest sensitivity, but not to 5-FU, FOLFOX and FOLFIRI at the *ED* proposed. These results are in agreement with the literature suggesting that first-line FOLFOXIRI chemotherapy leads to improved survival and efficacy of metastatic colorectal cancer patient outcomes in comparison to FOLFIRI or FOLFOX chemotherapy [22].

We also tested the response of MIA PaCa-2 cells and observed a high sensitivity to GEM and GEM/nab-P treatments. This analysis was confirmed by the efficacy data from metastatic pancreatic cancer patients treated with GEM/nab-P [23]. This experimental evidence suggests that the selected therapeutic equivalent dose (*ED* = 5) is effective in killing tumor cells and, in principle, is predictive of the best pharmacological treatment. In fact, we recommend using the equivalent *ED* = 5 dose in any co-clinical trial using zebrafish avatars of hepato-biliary-pancreatic and intestinal cancers.

We believe that our work represents a starting point for further research aimed at validating zebrafish avatars as a support for oncologists in clinical routines.

Potential applications are the evaluation of the disease prognosis and chemosensitivity assays for predicting the most effective chemotherapy scheme. In fact, we validated an approach consisting in the xenotransplantation of pieces of the patient tumor tissue, after surgery and histopathology screening. The aim was to obtain a model used to test the response of the patient’s tumor to various chemotherapy regimens, with an assessment in less than one week (Appendix A).

The xenotransplantation of cancer cells isolated and propagated from patient tumors in zebrafish is a more popular approach than the xenotransplantation of tissue fragments. Unfortunately, isolated cancer cells tend to lose both cell heterogeneity and the stromal contribution. Moreover, during isolation and adaptation, clones with a higher proliferation rate than that of the primary tumor are selected and thus are not representative of the cancer cell population [11].

For precision medicine and personalized medicine, the xenotransplantation of biopsy or surgical specimen fragments screened by the pathologist is therefore recommended to develop patient-derived xenografts in which the stromal counterpart and cancer cell heterogeneity are both preserved [24].

Our experiments were performed using fresh tumor tissue selected by an anatomopathologist to reduce the presence of necrotic tissue. However, we performed an extensive histological analysis to monitor cell engraftment in zebrafish avatars, as DiI or DiO stainings are not sufficient for the discrimination of live and labeled cells from debris.

Our data suggest that fresh tumor tissue transplanted into 2 dpf embryos can engraft and survive in the host, as highlighted by histological analyses showing typical cancer cell morphology (H&E staining, Appendix A) and a small fraction of pyknotic nuclei (Figure 5).

The xenograft is positive to anti-human IgG staining (Figure 6), confirming the presence of cancer cells rather than the engulfment of DiI-labeled cell debris due to the infiltration of the zebrafish immune system cells which could partially account for DiI staining [25]. The survival rate of the xenografted host was acceptable, at both 1 dpi (81%, *n* = 101) and at 2 dpi (68%, *n* = 101). We also detected the capacity of cancer cell extravasation and dissemination in distal tissues (Figure 6D). As the relative area at 2 dpi/2 hpi was fixed as the primary measure of the study, we performed the efficacy tests under the assumption that a statistically significant decrease in this measure with respect to the control group (no chemotherapy) is a hallmark of the chemotherapy response. Using the relative area, rather than its absolute value, reduces the variability of the stained area among embryos due to the different sizes of the xenografted fragments and the effect of the DiI or DiO cell debris.

In fact, we evaluated the tendency of the stained area to decrease over time as a consequence of the treatment, rather than its size. Specifically, we tested the chemosensitivity in 24 human tumor fragments taken from surgical specimens. Pieces of tumor tissue were microinjected into zebrafish embryos to create zebrafish avatars and treated with the *ED* of chemotherapy drugs. Interestingly, our experimental data showed a good agreement with observations registered in common clinical practice. In fact, for PDX of colon cancer (Figure 9A), we found that the chemotherapy treatment was more effective when a combination of drugs was used (FOLFOX, FOLFIRI and FOLFOXIRI) compared to the use of 5-FU alone [26]. The greater aggressiveness of pancreatic cancers associated with a lower response to chemotherapy compared to colon and gastric cancers may explain why we never observed a complete response in our experiments for this group of pancreatic cancer PDX (Figure 9B). For the group of PDX of gastric cancer (Figure 9C), we found an excellent response to FOLFIRI which can be considered an acceptable first-line treatment for advanced gastric cancers [27].

Interestingly, the use of zebrafish avatars highlights the different responses to different chemotherapy regimens on a single-patient basis (Figure 10). Further tests will be necessary to fully validate the zebrafish avatar proposed here as a clinical tool predictive of the most effective treatment for each patient. Future experiments will focus on enrolling a higher number of cases in order to correlate the chemosensitivity results obtained in animal trials with the response to the chemotherapy treatment observed in human trials.

## 4. Materials and Methods

### 4.1. Zebrafish Husbandry

Zebrafish *(Danio rerio)* were handled in compliance with local animal welfare regulations (authorization n. 99/2012-A, 19.04.2012; authorization for zebrafish breeding for scientific purposes released by the “Comune di Pisa” DN-16/43, 19/01/2015) and standard protocols approved by Italian Ministry of Public Health, in conformity with the Directive 2010/63/EU. Zebrafish fertilized eggs were obtained by natural mating of *wild-types of* fishes at our facilities and the developing embryos were staged in incubator at 28 °C according to Kimmel et al. [28]. Before any procedure, embryos were anesthetized in 0.02% tricaine.

### 4.2. Cell Culture, Staining and Microinjections

The HCT 116 human colorectal carcinoma cells were cultured in McCoy’s 5A Modified Medium supplemented with 10% fetal bovine serum (FBS), 100 U/mL penicillin and 100 μg/mL streptomycin. The Mia Paca-2 human pancreatic carcinoma cells were cultured in DMEM supplemented with 10% FBS, 100 U/mL penicillin and 100 μg/mL streptomycin. Cells were incubated at 37 °C with 5% of CO_2_ in humidified atmosphere. Cells were detached at 80% confluence with 0.25% (*w*/*v*) trypsin- 0.53 mM EDTA solution and stained with 10 μg/mL CM-Dil for 15 min at 37 °C followed by 15 min on ice in darkness. Cells were washed and centrifuged three times by D-PBS and resuspended in D-PBS supplemented with 10% FBS to a final concentration of 100 cells/nL. All the reagents were supplemented by Thermo Fisher Scientific, Waltham, MA. Dechorionated embryos at 2 days post fertilization were anesthetized and injected with four nanoliters of cells suspension in the left side of the yolk using a heat-pulled needle and the PV830 Pneumatic PicoPump microinjector. The embryos were incubated at 35°C and one hours after injection were screened with fluorescence microscope.

### 4.3. Human Tissue Preparation and Transplantation into Zebrafish Embryos

The clinical study was approved by the ethics committee on clinical testing in Tuscany (“Comitato Etico Regionale per la Sperimentazione Clinica della Toscana - sezione AREA VASTA NORD OVEST” 09/11/2017, prot n 70213). Human material from surgical resected specimens was obtained from the University Hospital of Pisa (Italy) after written informed consent of the patients and approval of local Ethical Committee. Tumor tissue screened by the histopathologist (from the histopathology unit, Cisanello facility) was washed three times with RPMI supplemented with 100 U/mL penicillin, 100 μg/mL streptomycin and 2.5 μg/mL Amphotericin and cut into small pieces (1–3 mm) using a scalp blade. The pieces were then transferred to a 5 mL tube, and stained with either 40 μg/mL DiO in D-PBS (in case of esophageal and gastric cancers) or 40 μg/mL CM-Dil in D-PBS (in case of hepato-biliary-pancreatic cancers and intestinal cancers). The tissue pieces were incubated for 15 min at 37 °C and 15 min in ice cube. Tissue pieces were then washed and centrifuged three times by D-PBS and resuspended in D-PBS supplemented with 10% FBS. For tissue transplantation we used the manual method proposed by Marques et al. 2009 [12]. In particular, before transplantation, small pieces of stained tissue were further disaggregated using Dumont forceps (No.5) into a relative size of 1/4 to 1/2 the size of the yolk. Tissue pieces with the correct size were transferred to 1% agarose disks in multiwell plates in which the 2 dpf embryos were laying, ready for transplantation. A glass transplantation needle was used to transfer the tissue into the yolk. The tissue was picked up, put on top of the yolk and then pushed inside. The yolk usually sealed itself and in the majority of embryos, the tumor remained in the yolk. After transplantation, embryos were incubated for 2 h at 35 °C, then embryos were checked for presence of tissue and incubated at 35 °C in E3 supplemented with 100 U/mL penicillin and 100 μg/mL streptomycin with the presence or absence of drugs for the following days in the respect of the treatment plan.

### 4.4. Anticancer Drugs Safety and Treatment Plan

Groups of 30 embryos (2 dpf) arrayed in multi-well plates were exposed to E3 supplemented with 100 U/mL penicillin and 100 μg/mL streptomycin unmodified (control) and modified with the chemotherapy drug at 35 °C for 24 h added with increasing concentrations (Appendix A).

The drugs were refreshed each day for the three days of treatment plan. Three days after treatment (3 dpt) zebrafish embryos were fixed in 4% paraformaldehyde in PBS at 4 °C overnight. After that, they were dehydrated with increasing concentration of ethanol, and analyzed by stereo microscope to evaluate the phenotype (normal, death, aberrant).

### 4.5. Microscopy and Efficacy Evaluation

Two hours post injection (2 hpi) zebrafish embryos xenotransplanted with cancer cell lines were anesthetized with 0.02% tricaine and positioned laterally, with the site of the implantation to the top. The embryos were imaged by fluorescence microscope and transferred to a 24-well plate (one embryo/well) containing chemotherapy compounds in E3 supplemented with 100 U/mL penicillin and 100 μg/mL streptomycin or E3 supplemented with 100 U/mL penicillin and 100 μg/mL streptomycin unmodified (control). All embryos were imaged every day during the time course of the treatment. The size of the tumor area was measured by using ImageJ.

### 4.6. Histopathology

At 2 dpi (4 dpf) the xenografted embryos were fixed in 4% paraformaldehyde for 1 h at room temperature, followed by paraffin embedding for hematoxylin & eosin staining or OCT embedding for fluorescent immunohistochemistry staining. Embryos were respectively sectioned with microtome or cryostat, along the sagittal plane at a thickness of 8 μm.

Histopathological analysis was performed on paraffin sections stained by hematoxylin & eosin (Merck KGaA, Darmstadt, Germany) and digitally imaged using Nikon Eclipse E600 microscope.

Cryostat sections were incubated with FITC anti-IgG Primary Antibody (Roche, Basel, Switzerland) and Hoechst 33342 counterstained. Digital images of the stained sections were generated using a Nikon A1 confocal microscope. Pyknotic cells was counted at 60 X magnification within the epifluorescence DAPI image.

### 4.7. Statistical Analysis

We used GraphPad Prism 7 as statistical analysis software. Data analysis was performed by ANOVA, followed by Bonferroni correction or Dunnett’s post-hoc test or *t*-test. Statistical significance was set to 5%.

## 5. Conclusions

A simple, reliant and cost-effective avatar model based on the use of zebrafish embryos have been validated; by crossing data from safety and efficacy studies, we found a basic formula for the estimation of the dose to be used for running co-clinical trials. PDX has been derived from 24 human tumor fragments taken from surgical specimen of hepato-biliary-pancreatic cancers and gastro-intestinal cancers (XenoZ, NCT03668418) and results from chemosensitivity assays are in good agreement with observations registered in clinical studies.

## Figures and Tables

**Figure 1 cancers-12-00677-f001:**
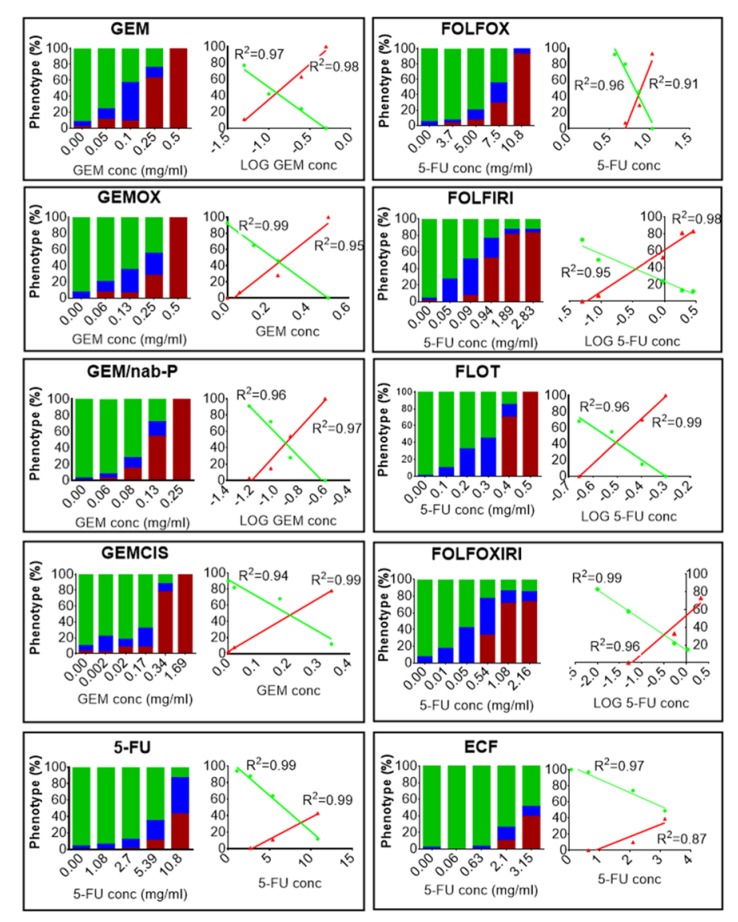
Chemotherapy safety study. Zebrafish 2 days post fertilization (dpf) embryos were incubated with media (E3 supplemented with 100 U/mL penicillin and 100 μg/mL streptomycin) modified with chemotherapy drugs or not modified at 35 °C for 3 days. At the end of the treatment, the percentages of dead embryos (red), aberrant (blue) and normal phenotype (green) were evaluated after fixation and stereomicroscope observation. In GEM, GEMOX, GEM/nab-P, GEMCIS, we report the Gemcitabine concentration in the x-axis (the concentrations of the other drugs are omitted but are provided in Appendix A). In 5-FU, FOLFOX, FOLFIRI, FLOT, FOLFOXIRI, ECF, we report the 5-Fluorouracil concentration in the *x*-axis (the concentrations of the other drugs are omitted but are provided in Appendix A). The control group showed a 10% alteration from the normal phenotype. For each chemotherapy regimen, a dose-response and the relative linear regression analysis of the normal phenotype (green line) and the dead embryos (red line) are shown. The resulting R square is reported. The results presented are a pool from three independent biological replicates (*n* = 90).

**Figure 2 cancers-12-00677-f002:**
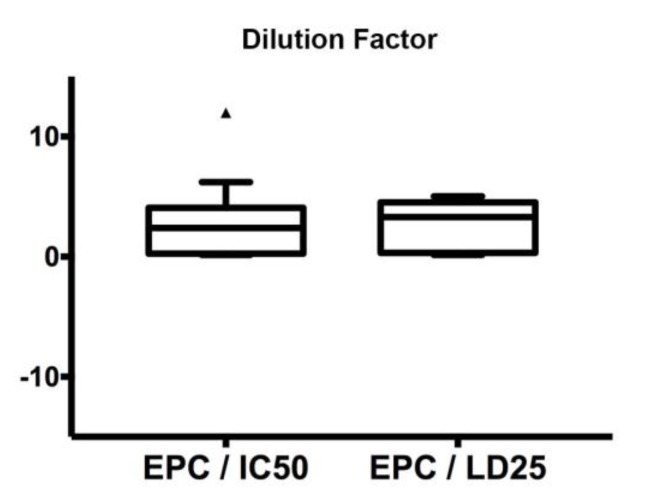
Estimation of the maximum tolerated dose (MTD). Box plot displaying equivalent plasma concentration EPC/IC50 and EPC/LD25 ratios for all chemotherapy protocols.

**Figure 3 cancers-12-00677-f003:**
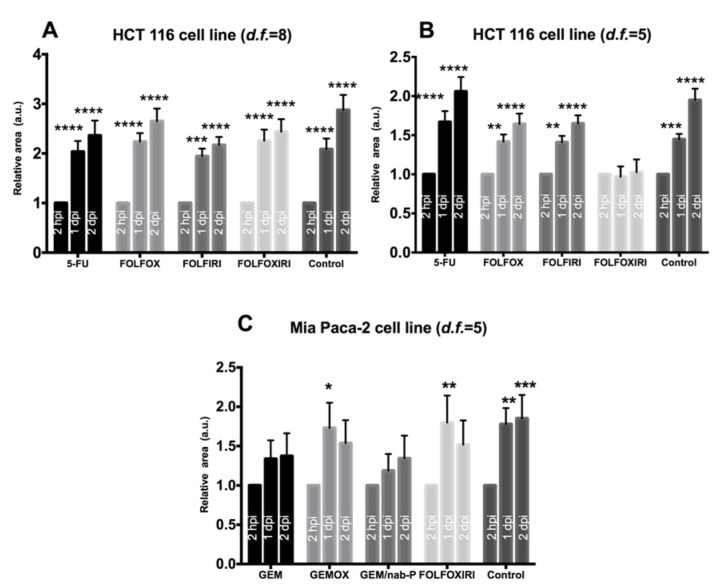
Efficacy analysis. Evaluation of the effects of chemotherapy on cancer cell lines (HCT 116, MIA PaCa-2) xenotransplanted in 2 dpf zebrafish embryos. Each embryo was imaged at 2 hpi, 1 dpi, 2 dpi and the relative area is the Dil-stained area normalized with respect to the 2 hpi time point. (**A**) Chemosensitivity of HCT 116 xenografts, *d.f.* = 8. A statistically significant increase in relative area was observed in all groups. (**B**) Chemosensitivity of HCT 116 xenografts, *d.f.* = 5. A statistically significant increase in the relative area was observed in the control, 5-FU, FOLFOX and FOLFIRI but not in FOLFOXIRI. (**C**) Chemosensitivity of MIA PaCa-2 xenografts, *d.f.* = 5. A statistically significant increase in relative area was observed in the control, GEMOX and FOLFOXIRI treatments but not in GEM and GEM/nab-P. Data are mean ± SEM and are representative of three independent assays. *N* = 15 (embryos), 2-way ANOVA followed by Bonferroni correction (all groups compared against control group). * *p* < 0.05; ** *p* < 0.01; *** *p* < 0.001; **** *p* < 0.0001.

**Figure 4 cancers-12-00677-f004:**
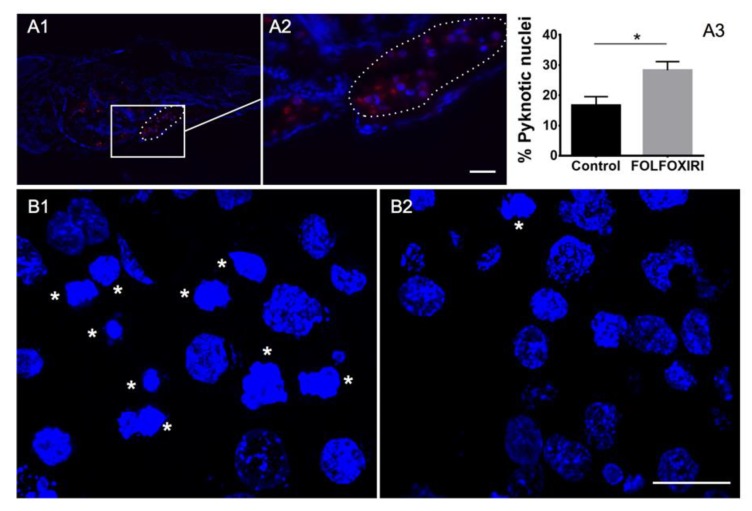
Cancer cell engraftment into zebrafish embryos 2 days after injection. (**A1**,**A2**) HCT 116 cancer cell line DiI labeled (red) injected into the yolk sac (circled) of zebrafish embryo 2 dpf. (**A3**) Quantification of pyknotic nuclei by HCT 116 cancer cells xenotransplanted after two days of FOLFOXIRI treatment vs. control. Data are mean ± SEM, *t*-test, *n* > 15 * *p* < 0.05. Sections from HCT 116 cancer cells xenotransplanted in a zebrafish embryo treated with FOLFOXIRI (**B1**) vs. control (**B2**). Stars: cells with pyknotic nuclei. Scale bar: 50 µm.

**Figure 5 cancers-12-00677-f005:**
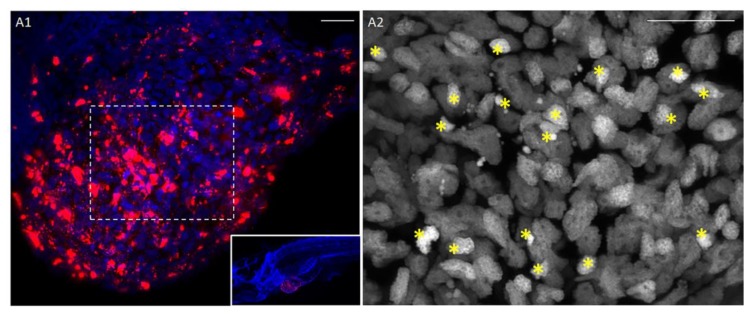
Patient-derived tumor xenografts. (**A1**) The fragment of a pancreatic cancer tissue (patient P053) injected into the yolk sac survives and invades the yolk sac. DiI (red) cell membrane staining, Hoechst (blue) nuclear counterstain at 2 dpi. (**A2**) Higher magnification of the dashed rectangle of panel B1 (Hoechst nuclear staining). Stars show pyknotic nuclei. Scale bar 20 µm.

**Figure 6 cancers-12-00677-f006:**
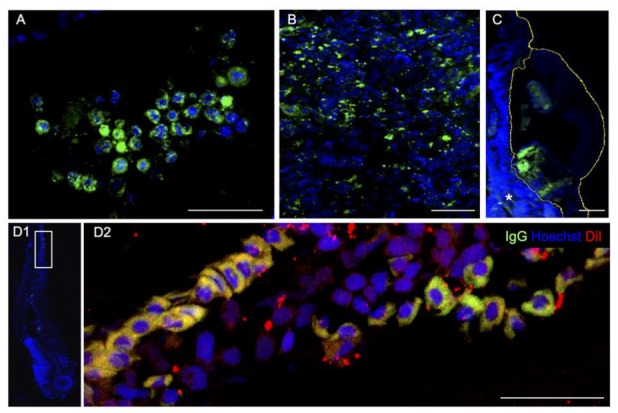
Anti-Human IgG immunohistochemistry (green). (**A**) HCT 116 cancer cell lines two days post injection into the zebrafish yolk. Patient’s pancreatic tumor before (**B**) and after xenotransplantation, 2 dpi (**C**). (**D1**,**D2**) Patient’s colon cancer cells spread throughout the vasculature reaching the zebrafish tail. Human cells are labeled with DiI (red) and anti-IgG (green). D1 is a magnification (90° rotation) of D2. Scale bar 50 μm.

**Figure 7 cancers-12-00677-f007:**
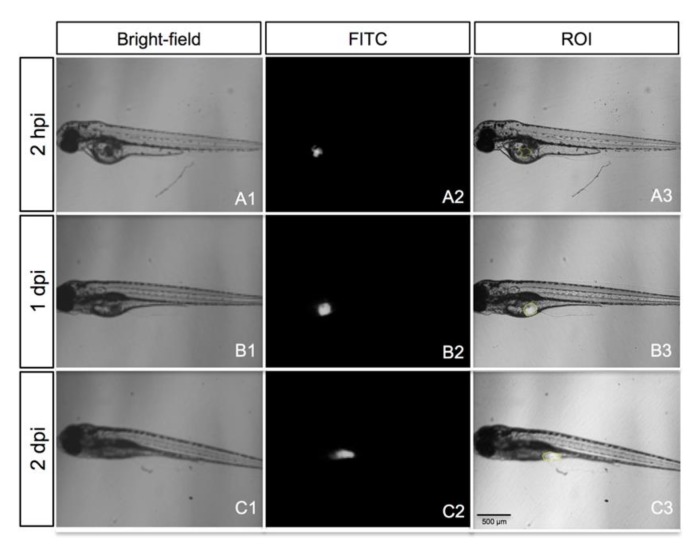
A representative embryo xenotransplanted with a fresh tumor specimen of gastric cancer (patient S013). Bright-field images of the grafted embryo (**A1**–**C1**), epi-fluorescence images (**A2**–**C2**) and overlay (**A3**–**C3**), showing the region of interest (ROI; yellow line). All images are oriented so that the rostral end is on the left and dorsal end is on the top.

**Figure 8 cancers-12-00677-f008:**
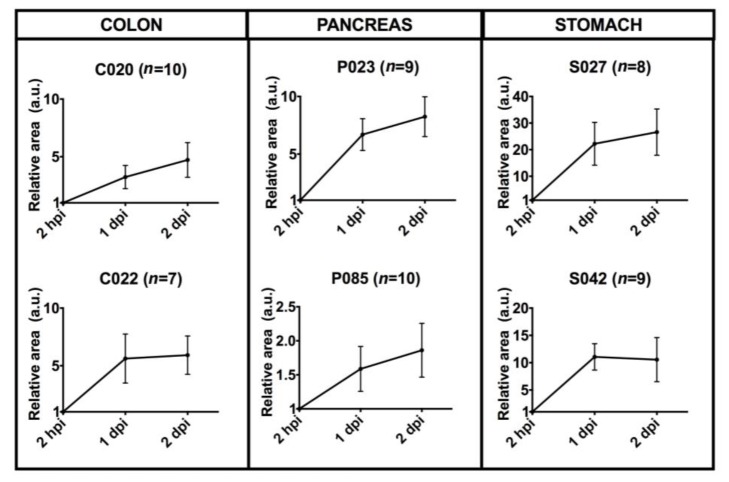
Quantitative analysis of six cases of patient-derived tumor xenografts. Dil-stained area at time points 2 hpi, 1 dpi and 2 dpi were normalized with respect to the time point 2 hpi. Patient enrollment codes are reported (C=Colon, P=Pancreas, S=Stomach), and the number of embryos analyzed for each case study is indicated in the image. Results are expressed as mean ± SEM. C020 (*p* = 0.04), C022 (*p* = 0.05), P023 (*p* = 0.003), S027 (*p* = 0.02), S042 (*p* = 0.02), P085 (*p* = 0.06) by 1-way repeated measures ANOVA.

**Figure 9 cancers-12-00677-f009:**
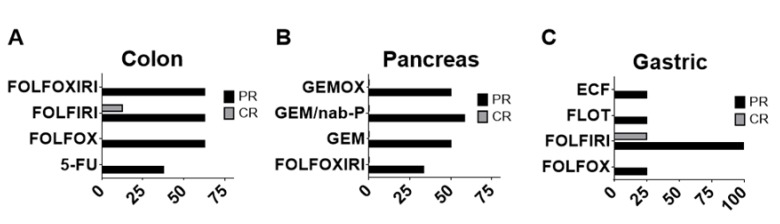
Percentage of partial response (PR) and complete response (CR). FOLFOXIRI, FOLFIRI, FOLFOX and 5-FU treatments in zebrafish avatars xenotransplanted with colon tumor (*n* = 8 patient samples analyzed) (**A**); GEMOX, GEM/nab-P, GEM, FOLFOXIRI treatments in zebrafish avatars xenotransplanted with pancreas tumor (*n* = 12 patient samples analyzed) (**B**); ECF, FLOT, FOLFIRI and FOLFOX treatments in zebrafish avatars xenotransplanted with gastric tumor (*n* = 4 patient samples analyzed) (**C**).

**Figure 10 cancers-12-00677-f010:**
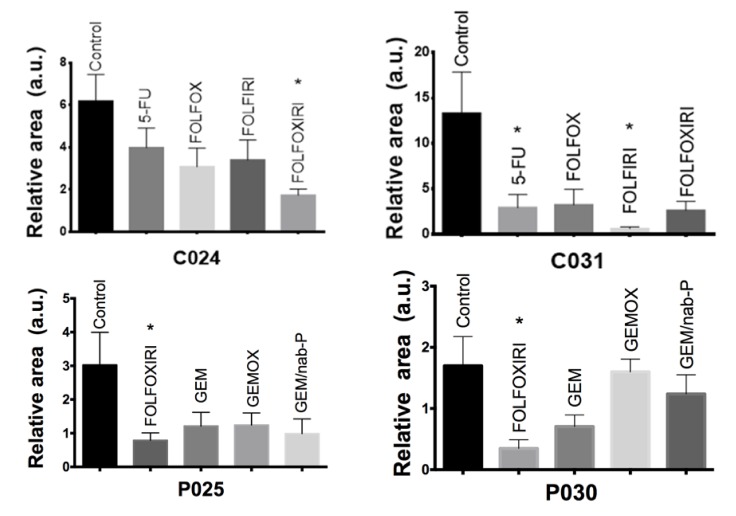
Chemosensitivity assay. Two dpf embryos were injected with fragments of patient tumor tissue and incubated for 48 h with chemotherapy compounds at the ED. Representative cases of colon cancer (C024 and C031 patient-derived xenograft (PDX)) and pancreatic cancer (P025 and P030 PDX) with quantitative analysis of the relative tumor area (2 dpi/2 hpi for colon and 2 dpi/1 dpi for pancreas). All graphs show an increase in the stained area over time in the control group. C024, P025, P030 show a statistically significant regression of the stained area size in the FOLFOXIRI treated group compared with control non-treated xenografts. C031 shows significant stained area reduction in 5-FU and FOLFIRI treated groups compared with control non-treated xenografts. Results are expressed as mean ± SEM and analyzed by 1-way ANOVA followed by Dunnett’s multiple comparisons test. * *p* < 0.05. C024: *n* = 5, 7, 8, 9, 5 and C031: *n* = 8, 7, 5, 5, 4, respectively for control, 5-FU, FOLFOX, FOLFIRI, FOLFOXIRI. P025: *n* = 9, 9, 8, 9, 7 and P030: *n* = 5, 5, 9, 3 respectively for control, FOLFOXIRI, GEM, GEMOX, GEM/nab-P.

**Table 1 cancers-12-00677-t001:** Chemotherapy protocols tested in the study, and corresponding equivalent plasma concentration (EPC) calculated as EPC = clinical dose x BSA / V, as BSA is the reference value of the Body Surface Area (1.8 m^2^) and V is the reference value of the blood volume (5.4 L). IC50, LD25 and relative ratio EPC/IC50, EPC/LD25.

Chemotherapy Protocol	Drugs Combination	Clinical Dose (mg/mq)	EPC (mg/mL)	IC50 (mg/mL)	LD25 (mg/mL)	EPC/IC50	EPC/LD25
**Gemcitabine**	Gemcitabine	1000	0.34	0.1	0.076	3.35	4.41
**GEMOX**	Gemcitabine	1000	0.34	0.23	0.16	1.50	2.13
	Oxaliplatin	100	0.034	0.023	0.016		
**GEM/*nab-*P**	Gemcitabine	1000	0.34	0.11	0.089	3.03	3.77
	*nab*-Paclitaxel	125	0.04	0.014	0.011		
**GEMCIS**	Gemcitabine	1000	0.34	0.19	0.1	1.75	3.35
	Cisplatin	25	0.01	0.0048	0.0025		
**5-FU**	5-Fluorouracil	3200	1.08	6.66	7.56	0.16	0.14
**FOLFOX**	5-Fluorouracil	2800	0.94	6.55	6.29	0.14	0.15
	Lederfolin	200	0.07	0.47	0.45		
	Oxaliplatin	85	0.03	0.2	0.19		
**FOLFIRI**	5-Fluorouracil	2800	0.94	0.15	0.19	6.21	5.03
	Lederfolin	200	0.07	0.011	0.013		
	Irinotecan	180	0.06	0.01	0.012		
**FOLFOXIRI**	5-Fluorouracil	3200	1.08	0.09	0.22	11.99	4.83
	Lederfolin	200	0.07	0.0056	0.014		
	Oxaliplatin	85	0.03	0.0024	0.0059		
	Irinotecan	165	0.06	0.0046	0.012		
**FLOT**	5-Fluorouracil	2600	0.88	0.28	0.27	3.08	3.23
	Lederfolin	200	0.07	0.022	0.02		
	Oxaliplatin	85	0.03	0.0093	0.0089		
	Docetaxel	50	0.02	0.0055	0.0052		
**ECF**	5-Fluorouracil	2800	0.94	3.29	2.54	0.29	0.37
	Cisplatin	60	0.02	0.07	0.054		
	Epirubicin	50	0.02	0.059	0.045		

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
