# Peer review of "A Model of a Zebrafish Avatar for Co-Clinical Trials"

_cancers, 2020, doi:10.3390/cancers12030677_

Round 1

Reviewer 1 Report

It is still not clear for the reviewer how the authors set EPC/IC50 and EPC/LD25 as presented in Figure2A. Since GEMOX, GEMICS, FOLFIRI and the others are regimens with the multi-drugs combination. Do the authors just calculated GEM and GEMICS based on GEM only? On the other hand, do the authors just calculated FOLFOX and FOLFIRI based on 5-FU? This is critical issue for the manuscript. Why the authors used so high 5-FU concentration used in toxicity assay for FOLFOX in comparison to FOLFIRI and FOLFOXIRI? The reviewer thinks that is the reason why EPC/IC50 and EPC/LD25 was so low in comparison to other 5-FU regimens in Fig 2A. The authors should explain the issue. On page 13 in line 339, “Figure 86A” should be Figure 8A

Reviewer 2 Report

The authors adequately adressed my comments.

Author Response

The paper has been revised by a native speaker

This manuscript is a resubmission of an earlier submission. The following is a list of the peer review reports and author responses from that submission.

Round 1

Reviewer 1 Report

The authors presented a potential clinical model of zebrafish avatar for colorectal, pancreatic and gastric cancer. This model is unique and useful since mouse avatar is sometime time-consuming and a high-cost model for clinical application. The manuscript is basically well-written and informative but there are some critiques before acceptance.

The authors should explain more precisely how they calculated EPC for multidrug combination chemotherapy. The authors showed that FOLFOXIRI was effective against C024 and C031 cells. The reviewer wants to know whether FOLFOXIRI was also effective against the patients. The authors should present whether zebrafish assay may reflect to chemotherapy outcome of a real patient treatment. In discussion session, the authors described that equivalent dose ED=5 will be applied in any co-clinical trial of zebrafish models. However, the authors only checked in colon cancer, pancreatic cancer and gastric cancer. Please explain why ED=5 will be used in other malignancies. ED might be different according to character of chemotherapy reagents. The size of the tumor area was measured by fluorescence microscope using DiO or CM-Dil. Do the dyes stain only living cells? The authors should describe the limitations of this model, that will give a critical information for the readers of the journal.

Reviewer 2 Report

Usai et al. aim to provide a general conversion factor to be able to convert human plasma concentrations of various chemotherapeutics for co-clinical trials in zebrafish avatars and they provide examples of co-clinical trials.

This is of great interest as zebrafish avatars could be an essential tool for personalized medicine.

However, the data presented in the current study is not yet fully supporting their derived conversion factor in my opinion and some additional explanation and experiments are needed.

Major points of criticism:

It is not entirely clear to me, how the authors derive the CFs. CF for EPC/LD25 is stated to be 4.1, but is this a mean value and if so, what is the deviation? The same holds true for EPC/IC50. It would be good to elaborate in detail on how the authors come up with this CF since this is one important conclusion of this manuscript.

In addition, the authors report quite some variability in their CFs (see Figure 2 A) in the range of almost 2 log folds (Fig.2 A EPC/IC50). Yet, the give a general CF >4.6. I think, a justification is needed here supporting this generalization.

For all the xenograft studies in zebrafish, the authors use DiI labeled human cells and their entire quantification is based on the area covered with DiI fluorescence. DiI is a lipophilic dye and still stains cell fragments, which can even be taken up by immune cells (macrophages). Hence, the authors need to show that what they are quantifying as tumor cells is actually living human cells. For this, they should perform antibody stains using human specific antibodies.

Minor points:

It would be good, if the authors could correlate their IC50 values in zebrafish with in vitro data as well. Ideally, conversion factors going from the dish to zebrafish and to patients would be great.

The authors introduce the lethal dose LD25, but then write LC25 in line 88.

In Figure 1 it is not clear from the description what the red and the green curves depict. This information should be added to the figure legend.

For the EPC of chemotherapeutic agents with multiple active compounds, e.g. GEMOX. How is the EPC calculated? Is it just 0.34mg/ml for gemcitabine as stated in S2 or is oxaliplatin (0.03mg/ml) added in this EPC? It would be good, if the authors could explain in more detail so that it is easier for the reader to follow their calculations.

Check usage of the term zebrafish larvae vs. embryos. It is not consistent and sometimes both terms are used in one sentence for animals of the same age (e.g. line 119).

The authors claim to quantify pyknotic nuclei as a readout for chemotherapy induced apoptosis. From Figure S1 it is not possible to easily see pyknotic nuclei. Furthermore, it is unclear from the figure legend, if A1/A2 show a zebrafish treated with chemotherapy or a control fish. It would be easier for the reader to show a representative example of both groups. Furthermore, the authors infer that any cell showing a bit of DiI label is a human cancer cell. They should provide evidence for this by doing an antibody staining using a human specific antibody for their target cell type.

Figure S3 B: The posterior part of the image seems to be taken with different settings (exposure?) as the background is more pronounced. If this is the case, the authors should indicate this as this is not common practice.

The authors need to show that the red dots in the tail are intact human cells, which have migrated, e.g. by performing immunostaining using a human specific antibody.

In Figure 5 A the authors should provide a y-axis label for all origins.
